Efficacy of computational predictions of the functional effect of idiosyncratic pharmacogenetic variants

McConnell Hannah 1
http://orcid.org/0000-0003-3922-6376 Andrews T. Daniel 1 dan.andrews@anu.edu.au
http://orcid.org/0000-0003-0788-6513 Field Matt A. 2 3 matt.field@anu.edu.au
1 John Curtin School of Medical Research, Australian National University , Canberra, ACT , Australia
2 Australian Institute of Tropical Health and Medicine, Centre for Tropical Bioinformatics and Molecular Biology, James Cook University , Smithfield , Australia
3 Immunogenomics Lab, Garvan Institute of Medical Research , Darlinghurst, NSW , Australia
Procter James
Electronic publication date: 2021 Jul 15
Publication date: 2021
Volume: 9
Electronic Location ID: e11774
Received 2020 Jul 1; Accepted 2021 Jun 23
Copyright: © 2021 McConnell et al.
Copyright year: 2021
Copyright holder: McConnell et al.
License: This is an open access article distributed under the terms of the Creative Commons Attribution License, which permits unrestricted use, distribution, reproduction and adaptation in any medium and for any purpose provided that it is properly attributed. For attribution, the original author(s), title, publication source (PeerJ) and either DOI or URL of the article must be cited.
License URL: https://creativecommons.org/licenses/by/4.0/

Keywords: Pharmacogenetics, Pharmacogenomics, Variant, Off-target, Missense mutation, Functional inference prediction

Funding: Australian Government NHMRC APP5121190 This work was supported by Australian government fellowship for Matt A Field: NHMRC APP5121190. The funders had no role in study design, data collection and analysis, decision to publish, or preparation of the manuscript.

==============================
Background

Pharmacogenetic variation is important to drug responses through diverse and complex mechanisms. Predictions of the functional impact of missense pharmacogenetic variants primarily rely on the degree of sequence conservation between species as a primary discriminator. However, idiosyncratic or off-target drug-variant interactions sometimes involve effects that are peripheral or accessory to the central systems in which a gene functions. Given the importance of sequence conservation to functional prediction tools—these idiosyncratic pharmacogenetic variants may violate the assumptions of predictive software commonly used to infer their effect.

Methods

Here we exhaustively assess the effectiveness of eleven missense mutation functional inference tools on all known pharmacogenetic missense variants contained in the Pharmacogenomics Knowledgebase (PharmGKB) repository. We categorize PharmGKB entries into sub-classes to catalog likely off-target interactions, such that we may compare predictions across different variant annotations.

Results

As previously demonstrated, functional inference tools perform variably across the complete set of PharmGKB variants, with large numbers of variants incorrectly classified as ‘benign’. However, we find substantial differences amongst PharmGKB variant sub-classes, particularly in variants known to cause off-target, type B adverse drug reactions, that are largely unrelated to the main pharmacological action of the drug. Specifically, variants associated with off-target effects (hence referred to as off-target variants) were most often incorrectly classified as ‘benign’. These results highlight the importance of understanding the underlying mechanism of pharmacogenetic variants and how variants associated with off-target effects will ultimately require new predictive algorithms.

Conclusion

In this work we demonstrate that functional inference tools perform poorly on pharmacogenetic variants, particularly on subsets enriched for variants causing off-target, type B adverse drug reactions. We describe how to identify variants associated with off-target effects within PharmGKB in order to generate a training set of variants that is needed to develop new algorithms specifically for this class of variant. Development of such tools will lead to more accurate functional predictions and pave the way for the increased wide-spread adoption of pharmacogenetics in clinical practice.

Introduction

Individual variability in drug response poses a large challenge to safe and effective patient treatment (Meyer, 2000; Pirmohamed, 2001). Many oncology drugs have be shown to be ineffective in subsets of patients, meaning that often multiple drugs must be tried before an effective treatment is found (Dancey et al., 2012). For example, it is not understood why statins (a class of drugs commonly prescribed for cardiovascular disease) behave differently between individuals (Silva et al., 2006), and can even cause a very severe toxic reaction in a small number of patients (Gabb et al., 2013). It has been estimated that 15–30% of this variability in drug response is due to genetic factors (Eichelbaum, Ingelman-Sundberg & Evans, 2006; Pang et al., 2009) however the precise mechanism of such genetic factors is often little understood. Numerous other factors consisting of both genetic and non-genetic components also play a role in variable drug response, including age, ethnicity, gender and differences in alcohol intake.

There are a growing number of databases that aggregate, curate and annotate the increasing body of identified genetic variants that occur in genes that interact with a pharmaceutical (pharmacogenes) (Sim, Altman & Ingelman-Sundberg, 2011). The Pharmacogenomic Knowledgebase (PharmGKB) (Whirl-Carrillo et al., 2012) is the largest, open database of pharmacogenetic data, and at time of publication, includes information on nearly 150 pathways and over 23,000 individual variant annotations. Variants within PharmGKB are also annotated with effect types (dosage, efficacy, toxicity) and the level of confidence (categories 1–4) of the pharmacogenetic association, with category 1 being the highest. The pharmacogenetic variants included in PharmGKB cover a wide range of mutation types, from nonsynonymous and synonymous single nucleotide variants (SNVs) to non-coding, intergenic and copy number variants.

Predicting the potential functional impact of a missense mutation is necessary, due to the disparity between the number of identified variants and the number that have experimentally-derived functional data. For missense mutations, this interpretation gap is presently filled by mutation function inference tools, such as PolyPhen2 (Adzhubei et al., 2010), CADD (Kircher et al., 2014) and SIFT (Sim et al., 2012). These functional inference prediction tools integrate sequence conservation and, often, structural information to predict whether alterations to the amino-acid sequence are likely to alter the function of a protein (Khan & Vihinen, 2010). These tools are known to suffer from high false positive rates with previous work yielding FPRs of >10–65% for 23 tools that were run across 4,880 confirmed ClinVar variants annotated as “pathogenic” or “benign” (Li et al., 2018). Generally, the algorithms work by deriving a multi-factorial score with higher values (with the exception of SIFT) representative of variants more likely to be damaging to the structure and function of the protein. Several algorithms bin their values into discrete named categories with PolyPhen2 applying labels of ‘benign’ for scores from 0 and <0.15, ‘possibly damaging’ for scores from 0.15 and <0.85 and ‘probably damaging’ for scores of 0.85–1. The algorithms are often trained on distinct sets of variants with CADD employing a machine learning model trained on a binary distinction between fixed variants arising since the human/chimp split and simulated de novo variants. The types of evidence employed by the algorithms are numerous with CADD for example considering 60 annotation sets in total. The annotation sets cover a wide variety of evidence types including sequence conservation (e.g. phastCons, GERP), epigenetic modifications (e.g. DNase-Seq, H3K9Ac), functional prediction (e.g. TF motif, amino acid change), and genetic content (e.g. GC content, CpG content) amongst others. Despite the diversity of evidence types however, sequence conservation remains the most highly predictive evidence type and to our knowledge all tools incorporate sequence conservation data in their algorithms. Hence, sequence conservation is important and widely used in classifying a variant as either benign or deleterious.

The reliance of such tools on sequence conservation is critical when considering pharmacogenetic variation. A recent study assessed the effectiveness of eight tools on variants in the RYR1 gene, which is linked to pharmacogenetic disorder malignant hyperthermia (MH) (Schiemann & Stowell, 2016). They compared MH-causative variants and common variants and found none of the prediction programmes could classify all variants correctly as either ’damaging’ or as ’benign’ respectively (84–100% range for sensitivity and 25–83% range of specificity). Specific missense mutations have been shown to cause adverse off-target effects with rs1050828 causing glucose-6-phosphate dehydrogenase (G6PD) deficiency which induces haemolytic anaemia from anti-malarial drugs such as primaquine (Gampio Gueye et al., 2019).

A broader study appraised mutation functional inference methods across a variety of pharmacogenetic missense variants and also found them to perform poorly with the effect attributed to the ill-suited training sets used to build the models on which the algorithms rely (Zhou et al., 2018b). Such studies led us to examine pharmacogenetic variants in order to identify subclasses that are likely to perform poorly with such tools, such as variants associated with adverse drug reactions (ADRs).

ADRs are broadly classified according to general mechanistic distinctions (Patton & Borshoff, 2018). Type A reactions are common and their effects are predictable and mostly dose-dependent. Type A reactions relate to interactions of a drug with its primary drug target. Conversely, type B reactions are less common and are mostly unrelated to the main pharmacological action of the drug. Type B reactions, sometimes also called idiosyncratic drug reactions (Uetrecht & Naisbitt, 2013), can be dose-dependent or dose-independent, may be immunologically-mediated and/or may involve off-target drug interactions (Patton & Borshoff, 2018). Immune-mediated Type B reactions involve the drug inducing a specific immune response, such as the development of a skin rash commonly caused by administration of penicillin (Weiss & Adkinson, 1988). Off-target drug effects can also occur without an immunological component, such as the interactions of anesthetics with the ryanodine receptor 1 (RYR1) protein causing malignant hyperthermia (Robinson et al., 2006).

We extracted all PharmGKB variants causing missense mutations and assessed the effectiveness of eleven functional inference tools. PharmGKB contains substantial numbers of variants, across all variant evidence levels, that are computationally predicted to be benign. We independently analyzed variant sub-classes associated with Type A and Type B reactions to determine whether the functional inferences differ. We find that most PharmGKB entries incorrectly classified as benign are generally off-target, idiosyncratic variants. As such, using current functional-effect prediction tools may produce poor inferences on idiosyncratic pharmacogenetic variants. Identifying lists of such variants generates a training set suitable to develop and calibrate new algorithms designed for this specific sub-class of variant.

Materials & methods

Pharmacogenetic variant datasets

A set of pharmacogenetic variants with dbSNP reference cluster identifiers (RS) (Sherry et al., 2001) were obtained from PharmGKB (Whirl-Carrillo et al., 2012) and custom overlap code used to combine variant annotations (Field et al., 2015; Waardenberg & Field, 2019). Variants within PharmGKB are classified by gene, type of effect, level of evidence, specific drug, chemical, disease and phenotype.

Classification of off-target pharmacogenetic variants

A simple classification scheme was devised to identify and confirm likely off-target variants (Table 1). Input variants for this classification were the complete set of PharmGKB variants (Whirl-Carrillo et al., 2012). First, of these variants, all synonymous and non-coding variants were excluded, leaving just missense variants. All clinical variants were then filtered for PharmGKB annotations of effect type ‘Toxicity/ADR’ for any particular chemical and/or drug. Variants were removed if they also had an additional effect type (other than ‘Toxicity/ADR’) for the same drug. Next, variants were removed if they were present in absorption, distribution, metabolism, and excretion process genes (ADME; categorized as such in the PharmaADME database; www.pharmaadme.org) or were annotated with Gene Ontology (24) categories of ‘xenobiotic metabolism process’ or with ‘transporter’. Lastly, variants not containing entries in ClinVar were removed.

Table 1 Classification criteria used to identify off-target pharmacogenetic variants from the PharmGKB database.

Step	Filter	Number variants	
1	Exclude synonymous and non-coding variants	561	
2	Include variants that have type:toxicity/ADR	339	
3	For drug and gene pairs, exclude variants with additional effect types other than Toxicity/ADR	273	
4	Gene containing variant is NOT an ADME process gene OR annotated in GO with ‘xenobiotic metabolic process’ OR ‘transporter’	196	
5*	Variant not containing an entry in clinVAR	142	
Note:

* For 30 variants used as a truth set, we performed an additional detailed literature search to definitively classify each variant as type A or type B.

Validation of classification scheme

To validate how effective this classification system was at capturing off-target variants, we randomly sampled high-confidence, missense variants from the PharmGKB (Whirl-Carrillo et al., 2012) until we derived 30 variants with a known pharmacogenetic mechanism (Table S1). For each of these 30 variants we conducted a detailed literature review to manually classify each variant as Type A or Type B. Note this is an additional step not described in Table 1. Manual classification was geared toward being stringent and followed a checklist where the variant was assigned to Type B if it satisfied all of the following criteria: (1) was not a metabolic process gene associated with normal metabolism of the drug, (2) was not in a gene associated with the system the drug is prescribed to target, and (3) did not have a dose effect. These 30 variants were taken as gold-standard data for benchmarking our approach to determine variants associated with off-target effects.

Functional effect prediction

For all PharmGKB missense mutations, the predicted functional effect of mutations with SIFT v5.2.2 (Sim et al., 2012), PolyPhen2 v2.2.2 (Adzhubei et al., 2010), CADD v1.5 (Kircher et al., 2014), DANN (pre-calculated results downloaded July 2019) (Quang, Chen & Xie, 2015), FATHMM v2.3 (Shihab et al., 2013), GERP++ v2 (Davydov et al., 2010), MutPred v2 (Li et al., 2009), Mutation Assessor v3 (Reva, Antipin & Sander, 2011), Mutation Taster v2 (Schwarz et al., 2014), REVEL v3 (Ioannidis et al., 2016) and PhastCons v1.5 (Siepel et al., 2005) were calculated relative to EnsEMBL canonical transcripts using dbNSFP v4.0a (Liu, Jian & Boerwinkle, 2013) and Variant Effect Predictor v94 (VEP) (McLaren et al., 2016) (Table S2). Instances where a tool produced no value for a given SNV were recorded as an NA value.

Receiver operator curves (ROC)

ROC curves were produced for the most-widely used subset of these tools (CADD, PolyPhen2, SIFT, Mutation Assessor, MutPred and REVEL) using the R package ROCR (Sing et al., 2005). All high-confidence PharmGKB variants (classified with the highest level of confidence ‘category 1’) were input as the positive control while a set of randomly selected common variants (MAF > 0.1) were input as the negative control using the Perl function rand() across the entire set of dbSNP v151 missense variants. The same analysis was next performed using all low confidence PharmGKB variants (classified with the lowest levels of confidence ‘category 3’ or ‘category 4’). Labels were inverted for SIFT due to lower scores representing likely damaging mutations and CADD and Mutation Assessor scores were scaled into the range of 0-1. Area under the curve and the Matthews Correlation Coefficient were calculated using the R-package ROCR. Code used in all analyses is available at https://github.com/mattmattmattmatt/pharmaco_paper.

Results

Distributions of pharmacogenetic variant functional inferences

Functional inference scores were obtained for 561 missense single nucleotide variants (SNVs) present in PharmGKB, that also had dbSNP cluster identifiers. Predictions were made for each SNV with eleven different prediction tools (SIFT, PolyPhen2, CADD, DANN, FATHMM, GERP++, MutPred, Mutation Assessor, Mutation Taster, REVEL and PhastCons) (Table S2). The distributions of scores from the most widely-used subset of these tools (CADD, PolyPhen2, SIFT, Mutation Assessor, Mutation Predictor and REVEL) are plotted with variants grouped by major PharmGKB category 1–4 (Fig. 1). The predictions calculated for these functional variants ranged widely from benign to deleterious. Four of the tools generate scores in the range of 0–1 (with 1 most damaging except SIFT 0 most damaging). Mutation Assessor and CADD employ a range of positive values with CADD calculating a Phred-quality score. For example, a CADD score of 20 implies the variant is ranked in the top 0.1% of all possible variant scores based on all possible changes in the human genome (CADD score of 10 is top 1%, CADD score of 20 is top 0.1%, CADD score of 30 is top 0.01%, etc.). For comparison to expected background levels, we also selected a random set of 2,155 common human missense SNVs with assigned RS cluster identifiers. Overall, the distribution of random variants mirrors our previous work with Polyphen2 exhibiting a characteristic hourglass curve with very few intermediate values (Miosge et al., 2015). These tools represent a broad range of methodologies available for mutation functional prediction and the categories of information used by each tool are annotated in Fig. 1 as seq (sequence conservation), struct (protein structural metrics), and ens (ensemble tool that integrates individual tools). The results demonstrate how some of the highest confidence PharmGKB variants annotated as functionally important are predicted to be benign. Of the 119 highest confidence category 1 variants, the majority are predicted to be deleterious by PolyPhen2 (median score 0.996), however five variants were classified benign (rs116855232, rs1057910, rs121909041, rs3745274 and rs1050828). The 183 variants in category 2 had a much broader range of predicted functional effects with 33 variants predicted as benign and an overall median score of 0.138, even less than the median score of 0.245 for the randomly selected variants. Similarly, the distribution of functional effect predictions in category 3 was strongly skewed towards benign variants (PolyPhen2 median score 0.012) and category 4 had a distribution very similar to the random variant set (PolyPhen2 median score 0.319). Category 1 variants were compared to category 2, 3 and 4 variants using a Mann–Whitney U test yielding p-values of 5.55e−07, 2.19e−24 and 1.03e−18 respectively.

Figure 1 Functional inference scores across all PharmGKB variants grouped by confidence level.

Distribution of functional effect scores of PharmGKB variants predicted by six mutation effect inference tools. Boxplots shown are of (A) CADD Phred score, (B) PolyPhen2 score, (C) SIFT score, (D) Mutation Assessor score, (E) MutPred score and (F) REVEL score. Scores are plotted for each tool in variant confidence categories (from 1 (highest) to 4 (lowest)) assigned by the PharmGKB annotation. Each tool is annotated with the information types it employs to make predictions—Seq: sequence conservation, Struct: protein structural metrics, Ens: an ensemble tool that integrates results of several individual tools. Each tool employs categorical cutoffs with the most damaging category colored as red.

The five category 1 variants predicted to be benign by Polyphen2 were examined in detail to examine whether these mutations act on the core protein function. rs116855232 is found to cause a Arg139Cys change in the Nudix hydrolase domain of NUDT15 which causes a loss of diphosphatase activity affecting the metabolism of Thiopurines. rs121909041 causes a missense mutation in CFTR and is associated with increased response to ivacaftor for the treatment of cystic fibrosis. rs1050828 causes a Val98Met mutation in G6PD, a gene which helps protect cells from oxidative damage. This mutation causes an adverse off-target reaction with the G6PD deficiency inducing haemolytic anaemia from anti-malarial drugs such as primaquine. The remaining two variants occur in the cytochrome p450 CYP2 family of enzymes which are important in a variety of physiological and toxicological processes. rs1057910 causes an Ile359Leu change in CYP2C9 and results in poor metabolism for a wide variety of drugs used in the treatment of diabetes, epilepsy, and cardiovascular disease. rs3745274 causes a Gln172His change in CYP2C6 and reduces response to various HIV infection treatments. These examples illustrate how in some cases the variants do not act on core protein function.

To better assess the performance of the individual algorithms on high confidence (category 1) pharmacogenetic variants, ROC plots were generated for CADD, PolyPhen2, SIFT, Mutation Assessor, Mutation Predictor and REVEL (Fig. 2). Area under the curve (AUC) and Matthews Correlation Coefficient were calculated (Table 2). Overall, Mutation Predictor had the highest AUC at 0.975 followed by REVEL at 0.942, PolyPhen2 at 0.852 and the remaining algorithms raging from 0.728–0.774. REVEL had the highest Matthews Correlation Coefficient at 0.794, followed by Mutation Predictor at 0.788 and the remaining algorithms raging from 0.321–0.489. We next performed the identical analysis on low confidence (category 3 and 4) variants and uniformly found lower AUCs ranging from 0.397 to 0.682. Overall, the AUCs are significantly different (p value 2.7e−4 using one-sided t-test with bonferroni correction for six subfamilies) between the high-confidence and low-confidence variants.

Figure 2 Receiver operator curves (ROC) for six functional inference tools vs Type A and Type B enriched pharmacogenetic variants.

Receiver operator curves (ROC) for CADD, Mutation Assessor, MutPred, PolyPhen2, REVEL and SIFT generated with ROCR. All high-confident PharmGKB Category 1 missense variants (Type A enriched) were input as the positive set while a set of randomly selected common variants (MAF > 0.1) were input as the negative set using the Perl function rand() across the entire set of dbSNP missense variants. The same analysis was also performed for all low-confidence PharmGKB Category 3 or 4 missense variants (Type B enriched). Labels were inverted for SIFT due to lower scores representing likely damaging mutations and CADD and Mutation Assessor scores were scaled into the range of 0–1.

Table 2 AUC and MCC from functional inference tools for PharmGKB category 1 (Type A enriched) vs PharmGKB category 3 and 4 (Type B enriched) pharmacogenetic variants.

Algorithm	Type A or Type B enriched	Area under curve (AUC)	Matthews correlation coefficient (MCC)	
PolyPhen2	Type A	0.852	0.489	
MutPred	Type A	0.975	0.788	
REVEL	Type A	0.942	0.794	
SIFT	Type A	0.774	0.358	
CADD	Type A	0.728	0.321	
Mutation assessor	Type A	0.763	0.396	
PolyPhen2	Type B	0.397	0.00338	
MutPred	Type B	0.682	0.300	
REVEL	Type B	0.498	0.106	
SIFT	Type B	0.410	−0.0400	
CADD	Type B	0.461	0.118	
Mutation assessor	Type B	0.427	0.0764	
Average Type A		0.839	0.524	
Average Type B		0.479	0.094	

In addition to missense mutations, tools such as CADD are able to generate scores for other variant types such as non-coding SNVs. While detailed analysis of this type of variant is beyond the scope of this study, we identified 14 PharmGKB high-quality category 1 and 2 non-coding variants and generated CADD scores. The median CADD score was 14.0, well below the average of 27.2 for category 1 PharmGKB and even less than the 22.1 for the random 2155 dbSNP variants.

Classification of variation in pharmacogenes to detect off-target effects

A prior study (Zhou et al., 2018b) demonstrated that functional prediction tools do not perform well across all pharmacogenetic variation. We hypothesized however, that the performance of the tools would differ across subtypes of pharmacogenetic variation, and that Type B pharmacogenetic variants associated with off-target effects in particular would often be predicted to be benign. To investigate this possibility, we devised a simple classification system to enrich for Type B variants from PharmGKB (described in “Materials and Methods” and summarized in Table 1). Our starting data was all possible PharmGKB variants found to cause missense mutations. In order to discern how effective this classification system was at capturing Type B variants, we randomly sampled from the starting set of PharmGKB variants until we derived 30 variants classified as category 1 or 2 from distinct genes with a published pharmacogenetic mechanism (Table S1). While not exhaustive, this manually-intensive process covered almost half of the 61 total genes in pharmGKB containing variants with evidence levels of either category 1 or 2. For each selected variant we performed a detailed literature review in order to determine whether each variant should be classified as Type A or Type B. Of the 30 variants, nine were determined to be Type B variants (30%). We next applied our simplified classification scheme (Table 1) to the same 30 variants to identify candidate Type B variants (Table S1) which identified eight putative variants. Of these eight, five were true positive Type B variants (with no false negatives) and three were false positives. Given this retention of Type B variants from the unfiltered variant pool (9/30 = 0.3) to the enriched pool (5/8 = 0.63), we estimate from this data that this classification scheme yields a 2.1-fold (0.63/0.3) enrichment of Type B missense variants, with a sensitivity of 63% (5/8) and a specificity of 100% (21/21). Subsequently, using our simplified classification scheme, we analysed the full set of 561 PharmGKB variants causing missense mutations yielding 142 candidate Type B variants (Table S3).

Discussion

In this work we have appraised whether pharmacogenetic variants associated with Type B, off-target effects are consistently predicted to be less deleterious than other functionally-important variants. We find this to be the case and postulate that this result arises from the reliance of the current generation of missense mutation inference tools on sequence conservation information. Generally, when a deeply conserved genetic element is found to be mutated, this will result in this mutation being predicted to be deleterious. However, should a nucleotide not be conserved across deep evolutionary distances, mutations of that nucleotide are likely to be predicted to be ‘benign’. Should that nucleotide code for an amino acid or otherwise directly interact with a drug which confers a life-saving benefit, however, this would clearly not be the case.

We further devised a simple classification scheme to divide pharmacogenetic variants into Type A or Type B. Classifying pharmacogenetic variants is important as unless a pharmacogenetic variant is related to the evolved functions of a gene, then no information is present in the ancestral sequence record to inform on predicted functional importance. Many genes that interact with drugs (pharmacogenes) contain variants which generate Type A ADRs. For these Type A pharmacovariants, the drug is just a xenobiotic compound which the target-protein acts upon. Variants which adversely affect the function of the pharmacogene should be correctly classified as ‘deleterious’ by the current generation of functional inference tools. This is supported by previous work showing an association between the residual evidence intolerance score (RVIS which measure the tolerance of a gene to mutations) and targets of approved drugs (Nelson et al., 2015), however it is unknown whether this holds for off-target variants. In our classification, we define Type B variants as those that have only Toxicity/ADR effects, and appear in genes not annotated as part of an ADME process, GO term “xenobiotic metabolic process” GO term “transporter”, but do have a known mechanism which has been implicated with an idiosyncratic drug/protein interaction. Variants that cause a type B or off-target effect are much less likely to be subject to the same selection pressures as those of type A, meaning they may be incorrectly classified as ‘benign’ due to the lack of observed sequence conservation. Indeed, we show most off-target pharmacogenetic variants of this type are predicted to be functionally unimportant and will be missed using current tools. Zhou et al. (2018a) similarly observed that the genes containing many pharmacogenetic variants are often poorly conserved, making the reliance of the algorithms on sequence conservation alone problematic. The quality of the multiple sequence alignments is also important with the class of multiple sequence alignment algorithm selected shown to substantially impact downstream analyses (Blackburne & Whelan, 2013). An additional consideration is the constraints imposed by domain structure on missense mutations across the human genome (MacGowan et al., 2017). MacGowan et al. identified regions of the genome depleted of missense mutations and while most such regions were conserved across species, they identified regions that are not conserved yet were enriched for pathogenic variant, ligands, and DNA and protein binding interactions. Such variants cannot be accurately predicted using tools that rely solely on sequence conservation and require additional information such as genome-wide missense depletion scores. Similarly pharmacogenetic variants that are not conserved across species will require additional information for existing tools or even new tools altogether to accurately predict their functional impact. Given the importance of pharmacogenetic variation and the numerous nature of Type B pharmacovariants, new methods are urgently needed to capture this important class of variation.

While sequence conservation is a useful metric for predicting the impact of many variants, we have shown for certain subclasses of variants it is not suitable. However, without using sequence conservation information as a primary discriminator, what methods and datasets are available to differentiate between truly benign and functionally important variation causing off-target effects? In hopes of finding new ways to predict damaging missense mutations, researchers are increasingly applying machine learning techniques to improve functional prediction algorithms particularly for identifying disease causing variants (Kalinin et al., 2018). However, for pharmacogenetic variants options are limited. A recent study reported improved sensitivity and specificity using a functionality prediction framework optimized for pharmacogenetic variants however no code has yet been released to independently assess this claim (Zhou et al., 2018b). Regardless of the eventual outcome, the ability to accurately predict pharmacogenetic variants associated with off-target effects is critical for the increased adoption of pharmacogenetics in clinical practice.

Conclusions

Pharmacogenetic missense variants represent a complex set of genetic factors with highly diverse functional mechanisms that influence drug efficacy. Here we assessed the performance of functional prediction tools (all of which rely to some degree on sequence conservation over deep evolutionary timescales) on different subsets of pharmacogenetic variants. Our analysis confirms that variant sets enriched for off-target, type B adverse drug reactions perform significantly worse than both randomly selected common variants (MAF > 0.1) and variants enriched for Type A reactions that relate to interactions of a drug with its’ primary drug target. We describe a simple method to identify candidate variants associated with type B reactions and note that the majority are predicted to be benign and functionally unimportant. Generating a subset of such variants will enable the development of urgently needed new methods that can accurately detect and predict the impact of all pharmacogenetic variation.

Supplemental Information

Supplemental Information 1 Thirty PharmGKB variants with a known pharmacogenetic mechanism used for scoring classification scheme.

Click here for additional data file.

Supplemental Information 2 Functional inference prediction scores for all 561 PharmGKB missense mutations and 2155 randomly selected common dbSNP variants.

Click here for additional data file.

Supplemental Information 3 Candidate Type B PharmGKB variants.

Functional inference prediction scores from 11 algorithms for all 561 PharmGKB missense mutations and 2155 randomly selected common dbSNP variants.

Click here for additional data file.

We thank the National Computational Infrastructure (Australia) for continued access to significant computation resources and technical expertise.

Additional Information and Declarations

Competing Interests

Author Contributions

Data Availability

The authors declare that they have no competing interests.

Hannah McConnell performed the experiments, analyzed the data, authored or reviewed drafts of the paper, and approved the final draft.

T. Daniel Andrews conceived and designed the experiments, performed the experiments, analyzed the data, prepared figures and/or tables, authored or reviewed drafts of the paper, and approved the final draft.

Matt A. Field conceived and designed the experiments, performed the experiments, analyzed the data, prepared figures and/or tables, authored or reviewed drafts of the paper, and approved the final draft.

The following information was supplied regarding data availability:

The raw data is available in the Supplemental Files.

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
