# Peer review of "Efficacy of computational predictions of the functional effect of idiosyncratic pharmacogenetic variants"

_PeerJ, doi:10.7717/peerj.11774_

## Round 0.1 · original submission · Major Revisions

Reviewer 1 notes the value of this work, but both reviewers highlight fundamental problems with arguments supporting the central finding of your manuscript - that existing (protein sequence alignment based) variant effect predictors are not effective for evaluating the impact of variants in pharmacogenes that may lead to cohort-specific ADRs. These critical aspects are:
- protein sequence alignment based variant effect predictors are only effective for evaluating the impact of mutations in conserved regions as a result of purifying selection
- the assumption that purifying selection in sites important for ADR have not yet 'had time to occur'

It is my opinion that arguments concerning the presence or lack thereof of selection are irrelevant to this work, which in reality highlights a segment of genomic variants that are very difficult to analyse in-silico with current tools. The manuscript presents a methodology that allows this segment to be more clearly identified, and may therefore help advance the development of more nuanced variant effect prediction tools.

In addition to the reviewers own comments and requests, I have provided an annotated PDF with specific comments echoing those of the reviewers and enumerate them below.

E1. Line 64: "Classifying newly predicted pharmacogenetic variants is a challenge as the number of actual variants is several orders of magnitude larger than the cases that are presently backed by experimentally-verified functional data."

This needs clarification - pharmacogenetic variant prediction is not done by PharmGKB - if this is truly the aim of this paper then you need to do something more than to just evaluate variant effect predictor tools. Otherwise, please revise to 'Classifying newly identified variants' - since what you describe in this paragraph still applies to the general case, and if necessary provide an additional statement to describe how pharmacogenetic variants differ from other clinically relevant variants.

E2. Line 80-82:
"The reliance of such tools on sequence conservation is critical when considering pharmacogenetic variation as in many instances such variants will not have been subject to purifying selection on an evolutionary timescale."

Both reviewers also consider this a problematic statement - and it is not supported by the rest of the paragraph. Zhou in particular describes the issue differently - suggesting that variants responsible for pharmacogenetic variation are often in genes poorly conserved, hence more difficult to predict the impact of variation purely by sequence conservation alone.

Here, you may find relevant a preprint from my own colleagues describing work specifically addressing the problem of assessing mutational impact in variable regions: https://www.biorxiv.org/content/10.1101/127050v2
(please note that I have only suggested my colleague's work because of its pertinence to this discussion, and will not base any editorial decision on whether or not you have chosen to acknowledge or cite my colleague's work).

From another perspective, R1 notes a paper reporting an observed correlation between RVIS and approved drug targets that may help refine the arguments in this paragraph.

E2.1. You may also wish to note here that an outcome of the work you describe may help to formally define a new training set useful for recalibrating these algorithms.

E3. Consistency of methodology for evaluating the identification of 'type B' variants
Line 181 states:
"All variants, including non-coding variants were scored with CADD"
In the description of your filtering procedure you say you exclude synonymous and non-coding variants. (line 129-130 and Table 1). Why are they now being discussed here (particularly in the analysis given in Line 189) ?

E4. Line 199-200 states:
"Further precision may be obtained through excluding potential off-target variants with a deleterious functional prediction."
If this is simply due to more of the variants predicted to be benign having stronger evidence supporting an off-target effect you should explicitly say this here.

E5. More generally (than E4), R1 suggests and R2 demands more rigor in the evaluation of the effectiveness of the procedure for identifying type B variants. A simple two-class statistical analysis of the classification methodology will help support this.

E6. Line 207-209:
"Protein sequence conserved between species implies that the function of the encoded protein was intolerant to mutations with any changes removed by purifying selection over evolutionary timescales."
This is far too broad a generalisation - *overall* conservation is different to regional conservation and has been widely discussed in the literature. Many essential genes exhibit variation in 'non-essential' regions.

E7. Line 211: "No information is present in the sequence record"
by this I can only presume you mean the ancestral sequence record ? This is not completely true. Many VEPS use structure-based measures - identifying mutations that impact structural interactions. These do not rely on the sequence record, although it is possible to argue that the underlying structure prediction tools do employ the sequence record to detect the structural similarities used to assess the impact of variation.

E8. Fully justify or avoid arguments relying on the presence/absence of selection" in the discussion.
E8.1. Line 212-214 is unjustified:
"Variants causing off-target and/or idiosyncratic reactions in particular are too recent a selective condition for evolutionary processes to impact the sequence content, even within the human genome."
- this statement has neither supporting evidence or supporting arguments - as highlighted by both reviewers.

E8.2. Line 215-217:
"We confirm this by extracting all pharmacogenetic variants with an off-target mechanism in PharmGKB and show that almost all such variants are incorrectly classified as benign."
- Your demonstration only shows poor performance, not that these variants are subject to 'selection'. R1 and R2 propose several additional factors that need to be considered when interpreting the performance of different predictors, not least being that there is bias inherent in ADR datasets because many drugs affect a smaller number of pathways to a greater or lesser degree.

E8.3. Line 230-232
"New methods that can detect pharmacogenetic variation that has not been subjected to purifying selection are urgently needed to capture this important type of pharmacogenetic variation."
This sentence should be deleted. The preceding one could be built on to highlight that there is a need to better tools, but what those tools need to capture more effectively needs to be more clearly communicated.

Reporting & Typographic comments
- Please ensure specific terms are properly described at first use (e.g. ADME should be first spelled out in full, and CADD's Phred score needs to be properly described as requested by R2, 'pharmacogenes' should also be concisely defined in the introduction).
- I have suggested grammatical/typographical revisions in the annotated PDF
- Please ensure all tables are provided as machine-readable csv/tsv files. PeerJ does not always require this but it is my opinion that for this work, TSV files would be beneficial for the community. In particular, the ADR mechanism reported for aspirin in Table S2 (provided as a word document) is not legible when viewed in certain versions of Word.

·

Basic reporting

This manuscript aims to assess the ability of current functional prediction algorithms to predict the severity of pharmacogenetic variants that are thought to be associated with off-target adverse drug reactions of drugs.

On the whole the language of the manuscript is good, in clear English with the topic well introduced.

However, I feel the title a touch imprecise. A generic variant can’t have off-target effects only a defined function/impact on protein and/or genetic traits. In a sense, each genetic variant is doing exactly what it is supposed to do. However, variants can be associated with increased/decreased off target adverse effects of drugs. I would suggest to revise the title to reflect this and be more precise.

Additionally variants are referred to as off-target variants throughout the manuscript, this probably needs tightening up, maybe define it at the beginning: “variants associated with off target effects (referred to as off-target variants)”.

Experimental design

The hypothesis is that many functional variation prediction algorithms rely on sequence conservation to predict variants that have a functional impact on a protein with the sequence conservation maintained by some purifying selection. However, because drugs have only been used over a short period of time, no purifying selection has acted on the drug target with regard to response to the drug and as such, the algorithms perform badly with respect to predicting non-benign function consequences of variants in drug targets. Moreover, the same hypothesis can be applied to off-targets of drugs, i.e. there has been no selection against increased off target effects of drugs and as such functional prediction algorithms are not accurate as their core training is not generalisable to this context.

The hypothesis has been clearly defined in the manuscript, however, I feel there are some things to address with it, particularly with the lack of selection at drug targets. Obviously it is true that drugs have only been used over a short period of time and not enough time for selection to act. However, it is unclear if evolution would act upon an individual’s drug response or adverse drug response, for this to happen, it would have to impair their ability to reproduce which would essentially mean killing them before reproductive age. Given that the majority of the drug consumption is in the middle age/elderly (with the exception of recreational drugs) this is probably not the case.

Moreover, this question touches upon “what makes a good drug target?” and amongst the many attributes, is the ability to elicit a large response to a relatively small drug dose. There is some evidence of an association between reduced residual evidence intolerance score (RVIS) (a metric of the tolerance of a gene to mutational perturbation) and targets of approved drugs (10.1038/ng.3314 page 857), however, it is unknown if this extends to off-targets. However, given this association, it is unlikely that drug targets are just random and that there may be some selection operating on them as small variations within them have the ability to manifest in relatively large changes in the phenotype of an organism.

To guide the reader, when defining the hypothesis, it might be worth tackling what some of these algorithms are actually attempting to predict (i.e. the definition of deleterious) and perhaps give an example where deleterious amino acid changes have given rise to adverse off target effects. This will give the reader some context as to why one might expect a functional prediction algorithm to have some predictive ability for off-target adverse effects.

The classification scheme (lines 121-134) for off-target variants was good although, I am I am not 100% sure why preference was given to variants where “or when a protein or cellular system related to the intended drug effect was involved” (line 133). Surely off target effects can effect different systems? It was good to see the ADME genes removed though. Whilst the rationale is obvious for the removal non-coding and synonymous variants that will not be covered by the prediction algorithms – it would be nice to acknowledge (either in the introduction or discussion) that functional variation is just one avenue for genetic variation associating with adverse effects. One can envisage that increased or decreased expression of a target or off-target gene/protein in a non-therapeutic tissue could also cause (off-target) off-tissue adverse effects.

Functional Effect Prediction (lines 136-141), a nice range of prediction algorithms were evaluated, I am not 100% familiar with them all but I am sure some must have tuning parameters? It would be essential in an evaluation manuscript such as this to give more detail of the parameters used, perhaps in a supplementary note.

Validity of the findings

Overall the results are quite straightforward, many of the tools performed similarly. However, the authors only present the results from 6 of the 11 tools and it is unclear why the remainder are not discussed – although they do appear in the supplementary table 2, this should be referenced from the main text and probably should be a csv (if allowed) as the computational tractability of word docs is limited). I also, noted that some of the predictions in supplementary table 2 gave a lot of missing data, this probably needs commenting or assessed in some way.

Something, I am unclear on, is the classifications of the off targets are limited to categories 1A, 1B, 2A, 2B but the predictions are across 4 categories, I am not sure how to interpret this and perhaps some explanation is needed to guide the reader through, are you predicting across all categories even when they have no possibility of being classed as “off-target”? It maybe that I have miss-interpreted something here.

The section of the results discussing category 3 variants needs to be appraised in the context of how many type B variants are in the database overall as the reader can not gauge if category 3 variants are enriched for type B variants. The methods probably need adjusting to include category three in the off target predictions (see paragraph above).

I feel some consideration of causality of the variants should be given also. In the discussion for example the phrase “variants causing off-target effects” is used. I would suggest that this is changed to associated with off target effects, as in many cases the causal nature will be ambiguous. If we consider the results in the general context of causality, is the probability of being causal associated with category? Is this is why the prediction tools predict better for the most robust category 1 variants? Where as category 3 variants have a lower prior probability of being causal for an off target effect but are marking (in linkage disequilibrium) with a causal variant? In that way they may be predictive of an off-target effect whilst not demonstrating any predicted functional impact.

Overall, I think the results suffer from never really knowing the denominator for any of the analysis and I would recommend some sort of formal enrichment/association analysis. Also, no reference is ever made to the distributions of the random predictions.

Additionally, many of the tools predict on the quantitative scale but are interpreted on a categorical scale (benign, deleterious etc...). It would be useful to guide the reader where these cut offs exist on the scale. Additionally some context on what is the predictive accuracy under the best possible conditions will also help evaluation of the results

Additional comments

Overall, I think the topic of this publication is relevant and under explored. The role of genetic variation in adverse effects is not well understood. Additionally, many people use off-the shelf predictive tools with out considering what the tool is really predicting. In that sense I think this manuscript highlights these important issues, however, in it’s current form I do not think it fully addresses them and I feel that if the authors can address the points in this review this manuscript could add to the knowledge in this area.

Reviewer 2 ·

Basic reporting

Minor comment - line 38: ‘predicative’ is used incorrectly and could be deleted or replaced with ‘predictive’.

In addition comments in experimental design section overlap with considerations here.

Experimental design

1) The introduction should include greater discussion of the diverse evidence bases used, for example in the CADD tool.

2) Considering the sentence at the end of the introduction “Hence, the strength of purifying selection to remove non-functional or poorly functional variants from the population is the strongest evidence presently used to classify a variant as either benign or deleterious.” This is not well substantiated and should be supported by empirical results, removed or appropriately rephrased. It is not rigorous to claim that selection is the strongest evidence for variant classification simply because this information is commonly used by the available algorithms.

3) The methods section requires more detailed information about the datasets used in this study, so that a clear description for each dataset is given in a way that is accessible to the reader. This should include (a) more detail to specify exactly how the randomly selected set of 2155 SNVs were obtained and (b) the number of off-target SNPs that were curated, these should be included for each row of table 1.

4) The Figure and associated discussion does not give any indication of the meaning of the scores or threshold values that may be applicable in the six tools evaluated. For example the data for CADD shows a range of scores from 0 to 40, it is difficult to interpret the Figure without showing a threshold. Please refer to https://www.ncbi.nlm.nih.gov/pmc/articles/PMC4980758/

Also relating to the consideration of thresholds, in the text the authors set a conservative threshold value (20) for CADD and then identify that a majority of the high-confidence variants fall below this threshold. Results for a lower threshold value should also be quoted, a value of 10 would seem appropriate in this context (CADD suggest values between 10 and 20; a value of 10 has been shown to perform best in the above article Fig S1A).

5) In order to enable effective comparison of the tools evaluated ROC curves should be presented with an AUC value and where threshold values can be determined standard metrics should be quoted including Matthew’s Correlation Coefficient, TPR and FPR.

6) The authors should describe in detail the 6 variants classified as benign by PolyPhen2 that were found in PharmGKB category 1.

7) The discussion of score distributions (e.g for PolyPhen2) around different variant categories should take into account the distribution of nsSNVs, rSNVs and ncSNVs in these categories – which may explain the variation observed.

8) One limitation of this study is restricting to missense SNPs, because a portion of pharmacogenetic variation is non-coding. It would be valuable to include some consideration of noncoding variants in this work.

Validity of the findings

9) A central assumption implicit in this study is that algorithms which predict SNV effects on protein function should also identify pharmacogenetic variation. This does not always hold - for example, a SNV could disrupt a protein-drug interaction without impairing the protein’s function in the cell. While algorithms that predict SNV effects on protein function typically examine the relationship of predicted deleterious SNVs to disease/drug responses, it does not hold that all ‘pharmacogenetic’ SNVs are deleterious to protein function. Discussion around the implications of this assumption and it’s relevance to the work presented should be included in the manuscript.

10) The claim that "variants causing off-target effects are unlikely to have been subject to purifying selection" (in contrast to on-target effects) requires substantiation or should be removed from the manuscript. It may be argued that the large majority of deleterious mutations have not been subject to purifying selection (because they would have been eliminated from the population). There is a subtle difference to the author's statement in the above sentence, compared to the idea that evolutionary constraint at sites that contribute to variation in off-target effects should differ to that for on-target effects -- which is perhaps the message that is intended. However the manuscript does not provide evidence for the difference in evolutionary constraint discussed above. There are also a number of confounding factors involved that the authors have not accounted for or even discussed, such as the multiple biases towards specific biological pathways in drug development efforts.

11) The statement “our investigation showed little difference between tools which use structural features and those which do not” is not well supported. In it’s current form, the study has not compared the other tools in sufficient detail in order to draw this conclusion. Nor is the specific statement well formed because ‘little difference’ is too vague. Specific metrics should be employed to enable a quantitative (and qualitative) evaluation of the differing tools; and the similarities and differences considered. Some standard approaches for this are identified in point (5) above.

Additional comments

While the work embarks upon an interesting area of study, there are significant flaws in it’s current form - including issues with the reasoning and insufficient evidence to support the claims made.

---

## Round 0.2 · Major Revisions

Firstly - I must apologise for the delay in this decision. As you will see from the responses below, both reviewers considered the revised manuscript to be an improvement, addressing a great many of their original concerns. However - reviewer 2 in particular identified a number of problems which must be addressed (including some from their previous review), and also highlights that a number of statements still remain that are not justified or supported by evidence in the paper or backed up by citation. As a consequence, I have marked this decision as major revisions rather than minor.

In the annotated PDF I have added comments, grammatical corrections and a number of suggested rewordings which may help you in further revising this manuscript. These address the following issues:

1. I agree with R2 that statements regarding the reliance of existing variant classification methods on sequence conservation need to be 'toned down', and have made suggestions as to how the discussion could be revised. For the purposes of this work I suggest you instead more clearly emphasise deficiencies in the methodology strictly related to pharamacogenes: ie do the predictors employ information that can discriminate between sites that are pharmacologically important (regardless of their degree of conservation and evolutionary history).

2. Reproducibility. Thank you for reformatting the tables as TSV. I also recommend submission of the scripts used for statistical analysis and figure generation. Please review PeerJ's instructions to authors regarding deposition of scripts and data in Zenodo, etc. Where third party tools are used, version numbers or (at the very least) dates of use are useful to identify which versions of Uniprot/Ensembl/PDB in pipelines such as VEP.

3. Clarity and structure - R2 points out certain subsections should be reorganised. I have highlighted sections of the text (e.g. lines 200-227 - a wordy description of the box plots in Figure 2 that precedes a description of the ROC plots presented in figure 1)

4. Figure and results presentation. The ROC analysis figures introduced in this new version require revision for visual clarity. I also note problems with Figure 2 (boxplots) that make it difficult to compare performance across different predictors for different classes. Rearrangement of the layouts and avoiding the use of colour are the simplest improvements but there are almost certainly other ways these data could be presented that would improve clarity: e.g. the use of violin plots (if there are sufficient data points). As noted by R2 in their previous review, threshold lines could also be added to help the reader judge for each tool how well they have performed.

5. In R2's 'Validity of the findings' they request use of Mann-Whitney tests to more rigorously evaluate the significance of observed differences in performance between type and type B variants across the different prediction tools. They also request you provide brief details of the 6 type A variants misclassified by PolyPhen (these could be useful as examples in the discussion to contrast the issues confounding prediction of the impact of type A variants compared with the more difficult problem of detecting Type B variants).

·

Basic reporting

no comment

Experimental design

no comment

Validity of the findings

no comment

Additional comments

It is my opinion that the revised manuscript is greatly improved and that the authors have done a good job at implementing the comments from my review. I am of the opinion that it meeting the publication criteria for PeerJ.

Reviewer 2 ·

Basic reporting

a) The manuscript needs careful attention to and modification of the language used to ensure scientific rigour and clarity. For example:
- In the abstract "these idiosyncratic pharmacogenetic variants very likely violate the assumptions of predictive software commonly used to infer their effect" seems overstated/imprecise. This should be reformulated accordingly, for example to quantify or tone down (e.g. "... may violate the assumptions ...").
- In the introduction lines 54-55: "Numerous other factors play a role in variable drug response, including age, ethnicity, gender and differences in alcohol intake." this sentence uses 'other' to contrast with genetic factors. However the above factors include a genetic component; the text should be reformulated to avoid this confusion.
- The wording "These are data tools that" on line 72 does not read well and should be reformulated.
- Methods lines 170-171: 'These variants were used as a truth set to measure the performance of the classification scheme.' would be better phrased along these lines: 'These 30 variants were taken as gold-standard data for benchmarking our approach to determine variants associated with off-target effects.'
- Introduction lines 84-5; " The types of evidence employed by the algorithms are numerous with CADD considered 60 annotation sets based largely based on conservation" requires rewording for clarity (key issues underlined).
- The authors have used a structured abstract but omitted to include a final "Discussion" or "Conclusions" section - this section should be provided.
- The full stop is misplaced in the text "(pharmacogenes). (Sim et al. 2011)".
- Line 258 "...our classification scheme (Supplementary Table S3)." Should include reference to Table 1 for clarity.

b) Introduction, lines 75-76: The authors cite their own work to support the notion that predicted deleterious variants had little impact on protein function. This statement over-interprets the results (in PNAS 2015;112:E5189-E5198), which examined phenotypes in vivo and included a case study on TP53 transcription. Therefore this work assessed function at the level of changing the attractor state of protein networks which have evolved for robustness to perturbations in individual components and so is not equivalent to examining function at the single protein level. Indeed, prediction tools define 'deleterious' in relation to disrupting the properties of an individual protein (e.g. binding, catalysis). Accordingly the claim of >40% false positive rate is not supported by the work that they have cited. The introduction should be reworked to account for this.

c) Introduction lines 87-92. The following sentences present oversimplifications which should be corrected "Despite the diversity of evidence types however, in almost all cases if a variant or mutation lies in a highly conserved region in a multi-species alignment of orthologous gene sequences, the variant will very likely be considered deleterious or damaging. Conversely, should the variant be broadly similar to existing sequence variation in this same alignment, the variant will be considered benign or functionally homologous."

d) The point "Hence, sequence conservation is the strongest evidence presently used to classify a variant as either benign or deleterious." is not adequately substantiated by the preceding text. The authors could make reference to feature weightings (in the algorithms considered) for substantiation. Alternatively, the point could be toned down to say that sequence conservation is an important and widely used feature in identification of deleterious variants.

e) Organisation of the manuscript needs attention. For example:
- The methods section refers to results (especially Figure 1, Supplemental Table S1 and Table 2) which clearly should be not presented in Methods (but should be given in the Results section).
- In the Results section, lines 199-200 the sentence: "Instances where a tool produced no value for a given SNV were recorded as an NA value" should be placed in methods.

f) Lines 255 to 269 are difficult to follow and require revising for clarity/rigour. For example " Of the starting set of 30 variants, the filtering scheme retained eight putative Type B variants. Of these eight, five were true positive Type B variants (with no false negatives) and three were false positives."

Experimental design

h) Methods, lines 185-190. "All high-confident PharmGKB Category 1 variants were input as the positive set while a set of randomly selected common variants (MAF > 0.1) were input as the negative set using the Perl function rand() across the entire set of dbSNP missense variants. Labels were inverted for SIFT due to lower scores representing likely damaging mutations and CADD and Mutation Assessor scores were scaled into the range of 0-1. Area under the curve and Matthew Correlation Coefficient were calculated using the R-package ROCR performance function".

This text is not sufficiently detailed to enable reproduction of the work. A definition of "high-confident" should be given, and the score threshold value used to determine MCC requires explanation (how is the threshold determined, or is the maximum MCC value quoted?). In addition there are issues with language (e.g. "Matthew" should be "Matthews").

h) I cannot understand how the scheme in Table 1 includes a literature review (ie manual) step, but the results differ to the manual literature review step taken for a further 30 variants. How do the authors explain this?

Validity of the findings

i) Results lines 217 to 227; the authors quote median values for the PolyPhen2 scores in different categories of variants (four categories). It would be helpful to include statistical evaluation of these differences. One possibility could be to use the Mann-Whitney test, comparing each of the categories 2-4 against category 1.

j) The ROC curves are only calculated for Category 1 variants -- and several of the tools perform extremely well (e.g. with AUC values of 0.97, 0.94 or MCC values of 0.99, 0.97). These results appear to contradict the point by the authors that the algorithms do not perform well on the type B 'idiosyncratic' variants - however this aspect is not resolved or acknowledged in the manuscript text. An analysis should be undertaken to specifically investigate the difference in prediction performance on the Type A vs Type B variants (or the 'Type B enriched') set for each of the six main algorithms evaluated and the text adjusted accordingly.

k) Lines 235-236 "While there are significant differences," requires substantiation with p-values and specific comparisons.

l) The evaluation of the putative Type B variants (lines 267 to 269) given for PolyPhred2 should be performed across all six primary algorithms evaluated and must include statistical comparisons (for example using the Mann-Whitney test, with appropriate FDR correction for the six comparisons to be made).

Additional comments

m) In general, the authors have substantially improved the manuscript, including some refocusing and addition of new results which have partially addressed the comments made. However significant issues remain. There is still insufficient data supporting the key point about performance on 'idiosyncratic' (type B) variants. Indeed, at face value, some of the new data presented shows that the methods perform extremely well.

n) The authors have not addressed the point raised in the earlier set of comments about discussing the variants that are misclassified by PolyPhen. These variants should be used as case studies to exemplify the points made in the manuscript - in particular to address the question of the extent to which they disrupt the core protein architecture/function. It is straightforward to assess this using public databases (including information in dbSNP and UniProt). For example, rs116855232 is a Arg139Cys mutation in the Nudix hydrolase domain of NUDT15 that affects metabolism of Thiopurines by causing loss of diphosphatase activity (see https://europepmc.org/article/MED/26878724). This investigation took me just 10 minutes to perform and identifies rs116855232 as a false negative. It does not seem onerous for the authors to carry out similar investigations of the other variants and to discuss them in a paragraph in the manuscript in order to inform the point about misprediction vs functional differences.

---

## Round 0.3 · Major Revisions

One reviewer assessed your revised manuscript, and has identified an error in the MCC analysis requested in the previous round and presented in Table 2. As you may be aware, MCC can be used as a proxy for AUC when comparing classification algorithms with a particular dataset, so one would expect good correlation with MCC's -1->+1 range and AUC's 0->1 range - but Table 2 clearly shows that MCC calculations are incorrect. This has caused the one remaining reviewer to question the overall validity of the analysis and results presented.

Since your analysis and conclusions do not fundamentally rely on the MCC this error does not affect them, but MCC is indeed a better metric than AUC for evaluating classifier performance on unbalanced data (as noted by the reviewer), and these additional statistics were requested by both reviewers in earlier rounds to provide robust support for your observations of variant effect predictor performance.

1. It is imperative therefore that the R script is fixed and results are carefully reviewed. Please also revise the way that you report these values - "maximum reported Matthews Correlation Coefficient" (line 197) is not a meaningful description of the way this statistic should be reported or used.

The reviewer also highlights:
i. inconsistencies in author attribution between the submission metadata and the manuscript.
ii. that there is apparently incomplete data in the two final lines of the S4 csv file.

Both these also need to be addressed. Furthermore, you may also wish consider ensuring the manuscript's stated title matches the submission and truly reflects the nature of the work.

2. I note in line 346-349 you quote a 2-sample t-test applied to data in Table 2 in support of your suggestion that 'structurally based predictors do not seem to perform better on Type B pharmacovariants'. This raised the following concerns:

i. I Strongly recommend rewriting this sentence for clarity, rather than the cumbersome wording currently in parentheses "features and those which do not (Table 2: average AUC for tools using structural information is 0.849 vs 0.825 for tools that do not; p-value=0.79 2-sample t-test).".

ii. It is not at all clear to me from table 2 how you've arrived a these averages. Values contributing to each average shown in the table should be denoted with a '1' or '2' superscript, and the average described and reported in the table legend.

iii. Statistical assessment by comparison of AUC is known to be confounded when interpreting the impact of additional descriptors in predictive models - ie, statistical tests on AUC differences lack sufficient power to detect borderline significant improvements. A more robust and widely accepted approach is to directly test for heterogeneity amongst the matched predictions via McNamar (for two) or Cochrane's Q test (3 or more predictors). Raschka provides a both practical and citable explanation of how to do this: https://sebastianraschka.com/blog/2018/model-evaluation-selection-part4.html (citable via the arxiv doi).

I therefore recommend you consider whether it is important to include (and thus also provide robust statistical support for) the statement regarding the performance of variant effect predictors which do/don't employ structural data.


3. In order for PeerJ to accept this work please also comply with PeerJ's guidelines regarding submission of code and data (https://peerj.com/about/author-instructions/#data-and-materials). Embedding R commands as lines in a table is not acceptable, however small the script. Instead, a recognised repository (figshare, Zenodo) should be used - which has the advantage of allowing both code and data to be easily downloaded and executed by others.

4. Colours in box plots. In previous rounds I have noted that use of colour detracts from the clarity of statistical plots in certain figures. I cautiously recognise the apparent utility of the shading of the different segments of the vertical 'Score' access in Figure 1, but am left completely bemused as to why the boxes of the various box plots for the different predictors are shaded a variety of different colours. Please make the following changes:
i. Employ just one of the three or more colours currently used, and only to shade the 'damaging/deleterious' extreme
ii. Boxplots should be rendered with black outlines with a white fill *only*.
iii. Boxplots should be rendered *after* axes and threshold lines to ensure they appear above those other graphics.

5. A number of typos and issues of clarity also remain:

line 120: extraneous apostrophe "its", not "its'"
line 160: extraneous full stop '.'.
line 270: 'classification of pharmacogenetic variation' lacks clarity. I suggest you consider 'Classification of variation in pharmacogenes' to make it clear exactly what this paragraph focuses on (rather than classification of the different kinds of pharmacogenes). Similarly, line 272 needs to be revised. As remarked in previous rounds, this methodology is an important contribution and warrants clear description.
line 294: Missing 'S' - should be 'Supplementary Table S4'
line 318. Missing word after 'we': "classification, we Type B variants are those that have only Toxicity/ADR effects, and appear in"
line 331. Missing '.' "MacGowan et al."

Figure 2 labelling: Polyphen should be Polyphen2 in the title of first column's ROC plot for Polyphen2.

Reviewer 2 ·

Basic reporting

please see below

Experimental design

please see below

Validity of the findings

please see below

Additional comments

The manuscript has been considerably improved. However, a number of important issues remain, noted below.

1. Authorship is inconsistent (equal contribution/corresponding authors information) between first and second appearance of title and abstract in ‘Manuscript to be reviewed’ PDF file.

2. Title should be revised for clarity, for example the word ‘of ‘ appears three times

3. Structured abstract, Results section says that functional inference tools perform poorly on the complete set of PharmGKB variants. This conflicts with the data reported in table 1, with MCC values that are extremely good (although flawed, please see below).

4. I have serious concerns about the validity of results presented, especially for the type B variants. For example Polyphen AUC is reported at 0.40, with MCC=1; SIFT has reported AUC=0.41 with MCC=0.99. The reported AUC values (and shapes of the curves) are incompatible with the values given for MCC. Therefore, I attempted to replicate the values for Polyphen using supplementary table S4 and supplementary table S2. I took all of the 'random' variants from supplementary Table S2 as the negative class, and all of the type B variants in supplementary table S4 as the positive class. I found a similar, although not identical, AUC value (0.43 vs the reported value of 0.40). The maximum MCC value was 0.027, in stark contrast to the reported maximum MCC value of 1. I have deduced that this discrepancy is a result of an error in the computer code used to calculate the MCC value with the ROCR package (from inspection of the code given in Supplementary data S3). Specifically, the x.values are being quoted rather than the y.values from the ROCR ‘performance’ object that holds the MCC results. Furthermore, the authors appear to be using an unbalanced gold standard dataset, which will skew the AUC calculation towards the larger class (the negative class). This issue portends further concerns about the reproducibility and validity of the study as a whole.

5. There are issues with the data provided. For example the final line of supplementary table S4 does not have data for the first three columns, although prediction scores are provided.

---

## Round 0.4 · accepted · Accept

Thank you for addressing our requests in your revised manuscript, and in the accompanying github repository. I have verified that the scripts allow reproduction of the observed data, and that the correct values are now used for evaluating performance.

I also note the following minor issues that must be addressed:

1. Statistical calculations not yet documented

line 261-262: "Overall, the AUCs are significantly different (p-value 0.00512) between the high-confidence and low-confidence variants." - a simple one-sided paired t-test yielded a p-value of 0.00005, with bonferroni correction for 6 subfamilies, this is corrected to 2.7e-4 - still not equivalent to the p-value quoted. Please check and report the method used for computing this p-value.

Please consider the following R code :

# t-test to evaluate significance of differences between
# area under the curve and MCC for the 6 predictors
# for type A and type B variants

# data taken from table 2

perf <- data.frame(row.names=c("auc_typeA","mcc_typeA","auc_typeB","mcc_typeB"))
perf["PolyPhen2"] = c(0.852, 0.489, 0.397, 0.00338)
perf["MutPred"] = c( 0.975, 0.788, 0.682, 0.300)
perf["REVEL"] = c( 0.942, 0.794, 0.498 ,0.106)
perf["SIFT"] = c( 0.774, 0.358, 0.410, -0.0400)
perf["CADD"] = c( 0.728, 0.321, 0.461, 0.118)
perf["MutationAssessor"] = c( 0.763, 0.396, 0.427, 0.0764)

# perform paired t-test (1-sided)
auc_tt<-t.test(x=t(perf[1,]),y=t(perf[3,]),pair=TRUE, alternative="greater")
mcc_tt<-t.test(x=t(perf[2,]),y=t(perf[4,]),pair=TRUE, alternative="greater")

# bonferroni adjusted difference between type a and type b over all 6 methods
print("AUC ttest (Bonf. Corrected)")
print(p.adjust(auc_tt$p.value, "bonferroni",6))
# MCC corrected p-value comes in at 0.004 - robustly confirming your observation
print("MCC ttest (Bonf. Corrected)")
print(p.adjust(mcc_tt$p.value, "bonferroni",6))
{end R code}

2. Typos:
i. Fix sentence in line 199-201 - "Area under the curve and the Matthews Correlation Coefficient were calculated using the R-package ROCR performance function. ".
This should be "Area under the curve and the Matthews Correlation Coefficient were calculated using the R-package ROCR."

ii. line 359 - insert 'of' in "tools (all [of] which rely on "

3. Please ensure the statistical analysis script 'pharmaco.R' includes appropriate library() statements:
library(ggplot2)
library(ROCR)